# CROCHETBENCH: CAN VISION-LANGUAGE MODELS MOVE FROM DESCRIBING TO DOING IN CROCHET DOMAIN?

## ABSTRACT

We present CrochetBench, a benchmark for evaluating the ability of multimodal large language models to perform fine-grained, low-level procedural reasoning in the domain of crochet. Unlike prior benchmarks that focus on high-level description or visual question answering, CrochetBench shifts the emphasis from *describing* to *doing*: models are required to recognize stitches, select structurally appropriate instructions, and generate compilable crochet procedures. We adopt the *CrochetPARADE DSL* as our intermediate representation, enabling structural validation and functional evaluation via execution. The benchmark covers tasks including stitch classification, instruction grounding, and both natural language and image-to-DSL translation. Across all tasks, performance sharply declines as the evaluation shifts from surface-level similarity to executable correctness, exposing limitations in long-range symbolic reasoning and 3D-aware procedural synthesis. CrochetBench offers a new lens for assessing procedural competence in multimodal models and highlights the gap between surface-level understanding and executable precision in real-world creative domains.

## 1 INTRODUCTION

Procedural crafts such as crochet present a distinctive frontier for multimodal learning. Unlike traditional captioning or recipe datasets (Li et al., 2024; Hu et al., 2022; Mohbat & Zaki, 2024), crochet patterns intertwine three interdependent modalities: (i) **structured symbolic language**, where stitch abbreviations and counts define a precise grammar of construction; (ii) **long-form natural language**, which provides contextual guidance such as materials and sizing; and (iii) **visual evidence**, including photographs of completed objects and motif diagrams. Success requires not just alignment across modalities but step-wise reasoning that preserves *procedural fidelity*, making the challenge closer to *program synthesis* than generic description.

Crochet also offers a unique testbed for **3D-aware reasoning**. Each stitch encodes both local geometry and global connectivity, forming a topological structure that must be preserved across steps. Generating or interpreting patterns thus demands reasoning over how sequential operations accumulate into volumetric form. In effect, crochet couples symbolic instruction following with embodied spatial reasoning, cultivating abilities essential for domains where language must ground into physical tasks.

Despite the rapid growth of multimodal benchmarks (Fu et al., 2025; Li et al., 2023a; Zhang et al., 2025; Yue et al., 2024), existing datasets have largely focused on description or grounding. COCO (Lin et al., 2014) catalyzed captioning research, TextCaps (Sidorov et al., 2020) extended it to text-in-the-wild, and Recipe1M (Marin et al., 2018) explored cross-modal cooking instructions. While recipes also involve multi-step procedures, validating correctness typically requires real-world execution, making large-scale evaluation slow and resource-intensive. Crochet, by contrast, provides a symbolic domain where outputs can be automatically verified through DSL compilation, enabling scalable and efficient study of step-wise reasoning. Yet these benchmarks stop short of testing whether models can follow symbolic grammars, respect numerical and spatial constraints, and produce outputs that are *executable*. Current systems can describe, but not reliably *do*.

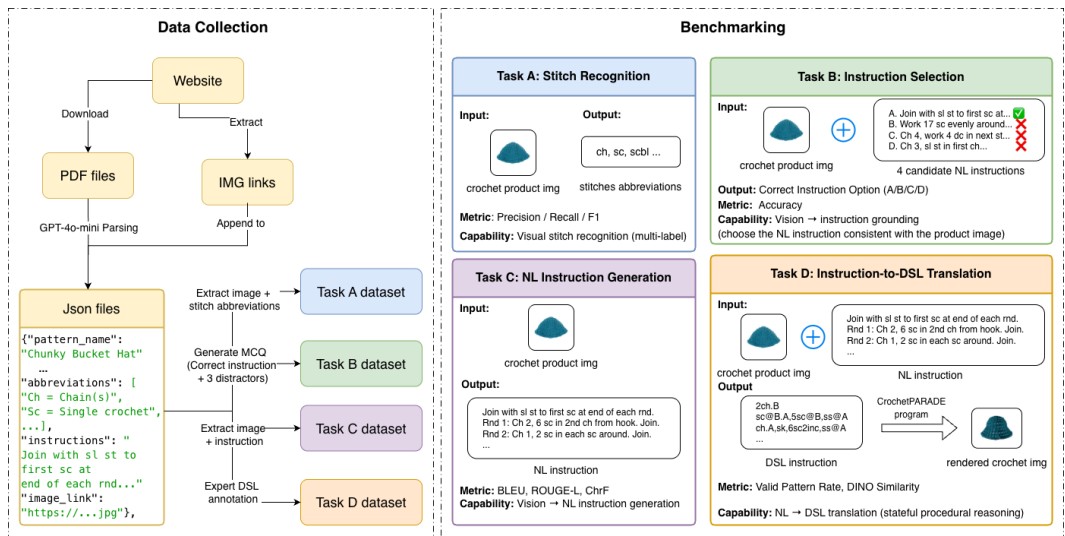

Figure 1: **End-to-end data construction and benchmarking workflow for CrochetBench.** The left panel illustrates the *data collection pipeline*: we download PDF files and image links from crochet pattern websites, and parse them using GPT-4o-mini to produce structured JSON files containing pattern metadata, stitch abbreviations, instructions, and product images. From each JSON record, we derive four supervised datasets: (A) stitch-level labels, (B) multiple-choice instruction selection items, (C) natural-language instruction generation pairs, and (D) expert-annotated DSL programs for procedural synthesis. The right panel summarizes the four benchmarking tasks: **Task A** evaluates multi-label visual stitch recognition; **Task B** measures vision-to-instruction grounding via MCQ selection; **Task C** assesses vision-conditioned natural-language instruction generation; and **Task D** tests stateful procedural reasoning via NL-to-DSL (Natrual language to Domain Specific Language) translation with execution-based metrics.

**CrochetBench** fills this gap by centering evaluation on **instructional fidelity**: can models not only recognize and generate, but also output step-wise, compilable instructions that respect symbolic, numerical, and topological structure? Each example in CrochetBench is a multimodal package—structured JSON metadata (stitch inventories and abbreviations), full-text procedures with rows/rounds and conditionals, and paired images of finished objects and motifs. Crucially, Crochet-Bench is paired with *CrochetPARADE* (Tassev, 2025), a domain-specific language (DSL) enabling executable evaluation, where natural language instructions are translated into compilable code enforcing geometric and topological coherence.

Our contributions are fourfold: (1) **CrochetBench**, the first executable benchmark for procedural textile crafts, unifying symbolic, textual, and visual modalities with evaluation protocols emphasizing procedural fidelity and 3D-aware reasoning; (2) a **comprehensive task suite** spanning recognition, comprehension, generation, and DSL translation; (3) integration of **CrochetPARADE into an executable pipeline**, enabling scalable, automated verification of outputs—unlike domains such as cooking, which require real-world execution—thereby shifting evaluation from surface similarity to procedural fidelity; and (4) **baseline analyses** of state-of-the-art VLMs/MLLMs, revealing systematic weaknesses including hallucinations, captioning bias, and structural artifacts.

## 2 RELATED WORK

Multimodal learning has traditionally focused on descriptive image–text pairs, such as COCO (Lin et al., 2014) and Flickr30k (Plummer et al., 2015). Recent benchmarks extend to procedural or instructional understanding, including Recipe1M+ (Marin et al., 2018) and large instructional video corpora such as YouCook2 and HowTo100M (Zhou et al., 2018; Miech et al., 2019). However, these tasks primarily evaluate semantic alignment or retrieval rather than whether a model can follow or generate a *correct* procedure. This gap motivates grounding multimodal evaluation in domains

where procedures are explicit, structured, and verifiable. To help readers unfamiliar with crochet, we provide a brief primer in Appendix A summarizing stitch types and pattern conventions.

Texile crafts provide precisely such a domain. Crochet patterns specify symbolic, stepwise procedures that determine the topology and geometry of a final physical artifact. Prior work in this area including Digital Crochet (Seitz et al., 2022) and Neural Inverse Knitting (Kaspar et al., 2019) demonstrates the feasibility of representing textile structures in machine-readable form but remains limited in scale and modality. By moving from general instructional data to a structured craft domain, we enable multimodal models to be evaluated on artifact-centric procedural reasoning rather than temporal action recognition. CrochetBench builds on this emerging direction by providing thousands of real crochet patterns with paired images and natural-language instructions.

To evaluate procedural correctness, CrochetBench adopts an executable domain-specific language (CrochetPARADE), linking our tasks to program synthesis benchmarks such as HumanEval (Chen, 2021), MBPP (Austin et al., 2021), and Spider (Yu et al., 2018). In multimodal settings, image-to-program benchmarks such as Im2LaTeX-100K (Deng et al., 2017) and pix2code (Beltramelli, 2018) similarly leverage executable formalisms for rendering-based evaluation. CrochetBench extends this executable perspective to textile crafts: patterns compile to structured instructions that can be rendered and validated, providing functional evaluation that tests whether a model's output *actually works*. This offers a lightweight alternative to domains such as chemistry or cooking, where validating a procedure requires physical or chemical experiments that are slow, costly, or impractical to scale.

## 3 DATASET DESCRIPTION

**CrochetBench** is a large-scale, structured benchmark comprising 6,085 crochet patterns across 55 distinct project categories. As shown in the left panel of Figure 1, the dataset is constructed from publicly available patterns on the Yarnspirations website[1], a widely used repository in the fiber-arts community. The raw patterns—originally distributed as PDF documents—were parsed and normalized through a GPT-4o-mini–based conversion pipeline that extracted and standardized key fields such as metadata, materials, measurements, gauge, abbreviations, and full step-by-step instructions. Each pattern was then transformed into a machine-readable JSON object following a consistent schema. Notably, 98.77% of patterns include an associated product image, enabling multimodal supervision for both recognition and generation tasks.

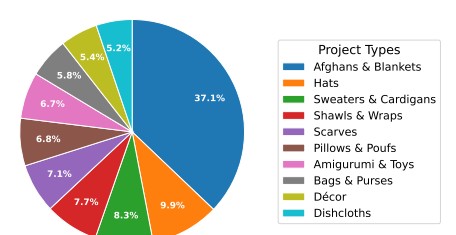

Figure 2: Distribution of the top-10 most common project types in **CrochetBench**.

The dataset supports diverse real-world crochet practices, with project types ranging from simple accessories to complex garments. Figure 2 lists the ten most common categories by frequency. The majority of patterns belong to a small number of dominant types—Afghans and Blankets alone account for over one-quarter of the dataset. More details can be found at Appendix B.

Each pattern is labeled with one of four primary skill levels, including *beginner*, *easy*, *intermediate*, or *experienced*. This allows for stratified evaluation across complexity tiers. Figure 3 shows the skill level distribution, which is strongly skewed toward beginner-friendly content. Only one pattern (0.02%) is missing a skill level label. More details can be found at Appendix B.

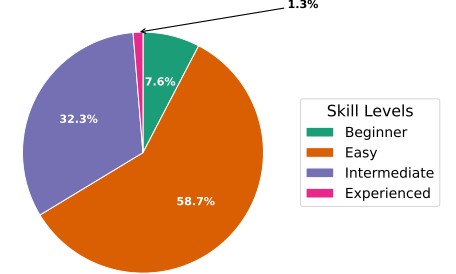

Figure 3: Skill level distribution across the **CrochetBench** dataset.

---

[1] https://www.yarnspirations.com/collections/patterns

Instructional complexity varies substantially across patterns. The number of characters in each instruction ranges from 20 to over 30,000, with a mean of 3,216 and a median of 2,453. Abbreviation counts (i.e., unique stitch tokens per pattern) range from 1 to 31, with an average of 10.6. These statistics are summarized in Appendix B.1. We observe a clear correlation between skill level and instruction length: beginner patterns tend to be short and use fewer abbreviations, while experienced patterns are significantly longer and more symbolically dense.

In addition to symbolic complexity, the dataset contains 3,143 abbreviation instances mapped to 789 unique standardized stitch tokens. This lexical mapping enables tasks such as vocabulary translation, sequence generation, and instruction validation. Beyond raw instructions, the structured schema also records rich metadata, including gauge, hook size, yarn weight, and measurements. A representative dataset entry is provided in Appendix 5.

Overall, CrochetBench provides a rich resource for multimodal modeling, symbolic reasoning, and structure-aware generation. Its coverage across diverse categories and complexity levels enables broad benchmarking of both open-ended generation and instruction fidelity tasks.

Table 1: Overall statistics of the CrochetBench dataset.

| | Total Patterns | Image Coverage | Avg. Instr. Length | #Project Types |
|---|---|---|---|---|
| **CrochetBench** | 6,085 | 98.77% | 3,216 characters | 55 |

## 4 TASKS

A central goal of CrochetBench is to evaluate whether multimodal LLMs can move beyond surface-level visual description and produce *procedurally correct* crochet instructions. Prior work shows that current models can describe crochet items (e.g., shape, color, texture) with high fluency, yet such descriptive competence does not imply an understanding of stitch structure or executable crafting procedures. To expose this gap, CrochetBench is organized as a progression of four tasks that isolate the core cognitive abilities required for real-world crochet reasoning, as summarized in Table 2.

Tasks A and B focus on **perception and comprehension**, representing the minimum prerequisites for procedural understanding. Stitch recognition and instruction selection evaluate whether models can ground visual cues in a structured stitch vocabulary and track local procedural dependencies within a pattern. However, identifying stitches or selecting a plausible next step does not guarantee the ability to synthesize a valid crochet procedure. Tasks C and D therefore target **procedural generation and formalization**, requiring models to produce coherent, stepwise natural-language instructions or executable CrochetPARADE programs. These tasks demand the integration of visual grounding, temporal consistency, symbolic manipulation, and domain-specific constraints. The following subsections describe each task in detail.

Table 2: Overview of benchmark tasks in CrochetBench. Tasks progress from recognition to comprehension, generation, and executable synthesis.

| ID | Ability Tested | Task | Evaluation Metrics | Test Size |
|---|---|---|---|---|
| A | Recognition | Stitch Recognition | F1, Precision, Recall | 6,009 (CrochetBench-A) |
| B | Comprehension | Instruction Selection | Accuracy | 6,003 (CrochetBench-B) |
| C | Generation | Instruction Generation | BLEU, ROUGE, ChrF | 6,009 (CrochetBench-C) |
| D | Formalization | Instr.-to-DSL (Step) | Valid Pattern Rate | 119 (CrochetBench-$D_{step}$) |
| | | Instr.-to-DSL (Project) | Valid Pattern Rate, Dino Similarity | 100 (CrochetBench-$D_{proj}$) |

## 4.1 TASK A: STITCH RECOGNITION

Task A evaluates a model's ability to identify crochet stitch types from an image of a finished product. We construct **CrochetBench-A**, a subset of 6,009 examples from the full benchmark, where each product image is paired with ground-truth stitch annotations. These labels are derived from the official pattern instructions and normalized into a standardized set of stitch abbreviations (e.g., `sc`, `hdc`, `dc`) to ensure consistency across patterns. Unlike standard image classification, this is a *multi-label prediction problem*: multiple stitches may co-occur within the same image, often with subtle visual differences in texture and geometry. This task therefore probes fine-grained visual grounding of structured crochet semantics.

**Evaluation.** For each example, we compute overlap between the predicted and reference stitch sets. True Positives (TP) are stitches correctly predicted; False Positives (FP) are stitches predicted but not in the reference; and False Negatives (FN) are stitches in the reference but missed by the model. From these counts, we compute precision (fraction of correct predictions among all predictions), recall (fraction of ground-truth stitches recovered), and F1 score (harmonic mean) Powers (2020). Metrics are averaged across examples to provide overall performance. This formulation rewards models that recover all present stitches while avoiding spurious predictions.

Accurate stitch recognition is foundational for the benchmark, as later tasks (e.g., instruction selection and instruction generation) depend on robust detection of stitch primitives.

## 4.2 TASK B: INSTRUCTION SELECTION

Task B evaluates whether a model can correctly associate an image of a finished crochet artifact with its corresponding natural-language instruction. We construct **CrochetBench-B**, a subset of 6,003 examples, where each instance contains one ground-truth instruction and three distractor instructions sampled from the same project category (e.g., hats, rugs). Because distractors originate from the same category, they share similar visual and lexical structure, thereby increasing task difficulty and preventing solutions based on superficial lexical overlap. The answer distribution across options is approximately uniform (A: 24.9%, B: 25.7%, C: 23.7%, D: 25.7%), ensuring no positional bias.

**Evaluation.** To support scalable and reproducible benchmarking, we formulate the task as a four-way multiple-choice question (MCQ). The model must select one option (A–D), with exactly one correct answer. Predictions are extracted using a deterministic regex-based parser that identifies explicit letter-based responses (e.g., "A", "Option B", "The answer is D"). Responses without a parsable choice are marked as unanswered. Accuracy is used as the evaluation metric.

This task provides a controlled measure of visual grounding and semantic alignment between images and procedural text, without requiring free-form generation. By forcing discrimination among near-neighbor instructions, Task B probes whether models can leverage fine-grained visual cues and domain-specific stitch semantics, which are essential precursors to reliable procedural instruction generation.

## 4.3 TASK C: INSTRUCTION GENERATION

Task C evaluates a model's ability to generate natural-language crochet instructions from an image of a finished item. We construct **CrochetBench-C**, a subset of 6,009 examples in which each image is paired with the corresponding ground-truth textual pattern. In contrast to captioning or stylistic description, this task requires generating a sequence of domain-specific commands (e.g., "Rnd 1: ch 4, 6 sc in ring"), each of which encodes precise stitch operations, counts, and ordering. Because real crochet patterns may include tens of steps, hierarchical structure (rounds, rows, substeps), and long-range dependencies, this task assesses whether models can infer the underlying procedural logic implied by the final visual product. The generated text must maintain consistent stitch semantics, preserve temporal ordering, and follow established formatting conventions used by human crafters.

**Evaluation.** We evaluate generation quality using BLEU, ROUGE-L, and ChrF (Papineni et al., 2002; Lin, 2004; Popović, 2015), which together capture complementary aspects of textual fidelity in procedural instructions. BLEU measures overlap of word-level n-grams and thus reflects local

lexical accuracy in stitch tokens and command sequences. ROUGE-L evaluates the longest common subsequence between the generated and reference patterns, capturing larger-scale ordering and structural alignment across multi-step procedures. ChrF operates on character-level n-grams, which makes it effective for crochet patterns where stitch abbreviations (e.g., `sc`, `sc2tog`) often differ by only a few characters. Word-based metrics treat such tokens as entirely distinct, whereas character-level comparisons can capture partial matches and small but semantically important variations.

However, textual overlap metrics alone cannot reveal whether the generated instructions form a coherent or executable procedure. A model may generate instructions that appear fluent and pattern-like while still violating fundamental structural constraints, including inconsistent stitch counts, infeasible transitions, or unbalanced repeat constructions. To directly assess structural correctness and program-level understanding, we introduce Task D, which requires models to formalize correct natural-language instructions into a machine-checkable DSL representation.

### 4.4 TASK D: INSTRUCTION-TO-DSL TRANSLATION

Tasks A–C evaluate perception, retrieval, and natural-language generation, but they do not test whether a model can represent crochet procedures in a structured, machine-interpretable form. Crochet patterns are inherently programmatic: they contain loops, repeats, and counting logic that natural language expresses only implicitly, and that text-based metrics cannot reliably validate. Task D isolates this structural dimension by requiring models to translate correct natural-language instructions into an executable DSL, thereby revealing whether models grasp the underlying program-like structure of crochet. This capability is essential for true procedural reasoning, and we instantiate it using the CROCHETPARADE DSL.

We construct two variants of Task D: **CrochetBench-D$_{step}$** (119 items) for step-level formalization and **CrochetBench-D$_{proj}$** (100 items) for project-level program synthesis.

**Step-Level Translation**    The step-level task evaluates whether a model can perform incremental NL→DSL translation, where "NL" refers to the natural-language crochet instructions written by human designers. Crochet patterns evolve step by step, and each instruction updates the underlying stitch state. Correctly translating a single step therefore requires maintaining consistency with all previous steps. In this setting, the model is provided with a prefix of correct NL–DSL pairs representing the portion of the pattern translated so far. Given the next natural-language instruction, the model must generate the corresponding DSL line. This formulation tests whether models can map local textual cues, such as stitch counts, increases/decreases, repeat structures, and turning logic, into the structured, symbolic operations of CrochetPARADE. Because crochet patterns are stateful, earlier context is essential for interpreting ambiguous constructs, ensuring round-to-round consistency, and encoding the correct update to the stitch topology. To capture variation in pattern progression, **CrochetBench-D$_{step}$** includes 52 early (steps 1–2), 34 mid (steps 3–4), and 33 late (steps 5–6) examples.

**Project-Level Translation**    In the project-level setting, the model is provided with the complete crochet instruction in natural language together with the corresponding product image, and must generate an entire CrochetPARADE program. This variant is globally self-contained but considerably more challenging than the step-level task: models must track stitch states over long horizons, resolve ambiguities in natural language, and produce code that is both syntactically valid and semantically aligned with the final design. This setting reflects how crochet instructions are used in practice, where each step depends on the correctness of all preceding steps. Image grounding is especially helpful for interpreting repeated motifs, symmetry, shaping, and termination conditions that may be under-specified in text alone.

**Evaluation.**    Because crochet patterns are inherently free-form—where multiple distinct programs can yield the same final product and a single natural-language instruction may admit several semantically equivalent DSL realizations—there is no canonical gold program for Task D. Exact string matching would therefore misjudge many correct solutions. Instead, CrochetBench evaluates correctness through *functional executability* using the CROCHETPARADE validator, which checks whether a predicted DSL program is syntactically valid, structurally consistent, and fully executable.

We use two complementary evaluation settings. For step-level translation, we report the **Valid Pattern Rate**, defined as the proportion of generated DSL steps that successfully compile. For project-level translation, we compute the **Valid Pattern Rate** for full programs and, for those that compile, render the executable portion into a crochet-like image and compute its **DINO Similarity** (Oquab et al., 2023) to the ground-truth product image, providing a coarse measure of semantic fidelity beyond syntax. To diagnose failure modes, we further identify the **first point of failure** for each invalid prediction and categorize it using our fine-grained error taxonomy (Appendix E), enabling us to distinguish local symbolic errors from broader state-tracking failures or misinterpretations of the natural-language instruction.

Table 3: Combined evaluation results across all three CrochetBench tasks: *Stitch Recognition*, *Instruction Selection*, and *Instruction Generation*. Best values are **bold**; second-best are underlined.

| | Model | Size | Stitch Recognition (%) | | | Instr. Sel. (%) | Instr. Gen. (%) | | |
| | | | **Prec** | **Rec** | **F1** | **Acc** | **BLEU** | **R-L** | **ChrF** |
|---|---|---|---|---|---|---|---|---|---|
| Open Source | BLIP-2 Flan-T5 XL | 3B | 29.53 | 23.03 | 22.50 | 25.62 | 0.21 | 9.26 | 9.32 |
| | Google Gemma 3 | 4B | 20.54 | 10.21 | 12.65 | 24.94 | 0.10 | 3.29 | 5.17 |
| | Google Gemma 3 | 27B | 17.19 | 18.14 | 16.05 | 24.94 | 0.40 | 5.17 | 6.55 |
| | DeepSeek-VL | 7B | 54.47 | **74.76** | 60.60 | 28.92 | 1.33 | 19.68 | 18.12 |
| | Qwen2-VL | 7B | 54.14 | 69.74 | 58.16 | 41.96 | 1.60 | 20.84 | 15.76 |
| | Qwen2-VL | 72B | 71.86 | 42.68 | 50.19 | **68.85** | 2.25 | 21.43 | 19.82 |
| Closed Source | GPT-4o | – | 62.14 | 59.39 | 58.01 | 58.11 | 3.33 | 23.53 | 23.80 |
| | Gemini 2.5 Flash-Lite | – | 74.49 | 49.77 | 56.83 | 55.63 | **4.82** | **25.83** | **30.20** |
| | Claude Sonnet 4 | – | **78.61** | 53.12 | **60.94** | 57.39 | 3.31 | 25.16 | 22.95 |

## 5 EXPERIMENTS

We evaluate a representative set of widely used vision–language models spanning open and closed ecosystems. For open source models, we include BLIP-2 Flan-T5 XL (Li et al., 2023b), Google Gemma 3 (4B and 27B) (Team et al., 2024), DeepSeek-VL 7B (Lu et al., 2024), and Qwen2-VL (7B and 72B) (Wang et al., 2024), covering a range of architectures and parameter scales. For closed source models, we evaluate GPT-4o (Hurst et al., 2024) , Gemini 2.5 Flash-Lite (Comanici et al., 2025), and Claude Sonnet 4 (Anthropic, 2025), which represent the strongest publicly accessible multimodal systems. These models span diverse architectures and parameter scales, providing a diverse and meaningful basis for assessing current multimodal capabilities on perception, retrieval, and procedural reasoning tasks.

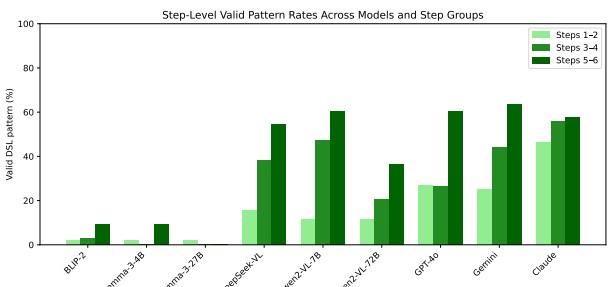

Figure 5: **Task D step-level translation** results showing the proportion of generated DSL lines that successfully compile for early (Steps 1–2), middle (Steps 3–4), and late (Steps 5–6) stages of crochet patterns. Across all models, valid pattern rates increase as more context is provided, but overall accuracy remains low. Even the strongest models struggle in early steps, indicating difficulty establishing correct stitch state and structural dependencies. Larger models (e.g., Qwen2-VL-72B and Gemma-27B) do not consistently outperform their smaller counterparts, highlighting that scale alone does not improve program-level structural reasoning.

**Perception and grounding improve with scale, but procedural generation collapses.** Table 3 summarizes results across Stitch Recognition (Task A), Instruction Selection (Task B), and Instruction Generation (Task C). Closed-source models achieve the strongest recognition performance, with Claude Sonnet 4 obtaining the highest F1 score (60.94%), and Qwen2-VL 72B leading among open models (50.19%). Although

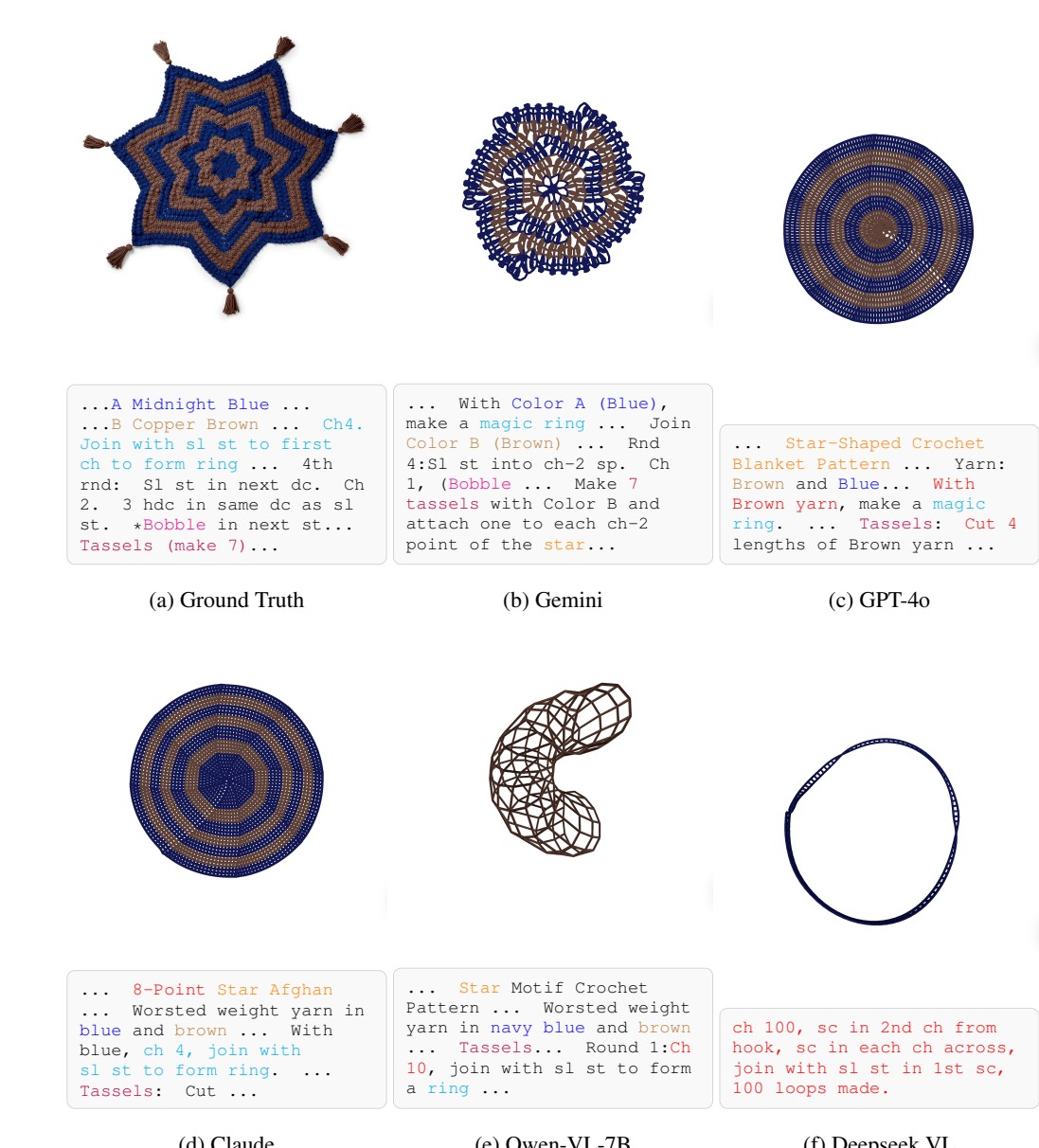

Figure 4: **Case study for Task C: Instruction Generation.** Each row shows the DSL-rendered output generated from the model's natural-language instructions and the color-coded instruction extract below it. Matching colors denote semantically corresponding elements across the reference and model outputs, while red marks incorrect or hallucinated steps. The ground truth is a seven-point star with alternating blue and brown yarn and tassels attached at each point. Gemini and GPT-4o generate structured and mostly coherent instructions but misconstruct the global geometry, producing a circular motif rather than a star. Claude and Qwen2-VL-7B misinterpret the shape more severely, producing circular or distorted wireframe-like forms. DeepSeek-VL collapses entirely into a degenerate single-loop pattern. Gemini is the only model to explicitly recognize the motif as a seven-point star, but its instructions still fail to produce the correct star topology.

larger models capture more fine-grained visual cues, accuracy remains far from saturated, and Instruction Selection shows similarly limited progress: Qwen2-VL 72B reaches 68.85%, while GPT-4o and Claude perform in the mid–50s, indicating that visual–textual alignment still depends on shallow correlations rather than robust grounding. These limitations become dramatically more pronounced in Task C. Natural-language instruction generation remains extremely challenging for all model, with BLEU, ROUGE-L, and ChrF scores uniformly low; even the strongest system, Gemini

2.5 Flash-Lite, achieves only 4.82 BLEU and 30.20 ChrF. The sharp drop from Tasks A–B to C shows that models capable of recognizing stitches or retrieving plausible text still fail to synthesize coherent multi-step procedures, reflecting fundamental gaps in procedural reasoning, symbolic consistency, and pattern-structure understanding.

**Surface-level fluency does not imply procedural correctness.** To better understand why instruction generation fails despite moderate performance on recognition and retrieval, Figure 4 presents a case study comparing model-generated natural-language instructions with their corresponding DSL renderings. Qwen2-VL-7B and DeepSeek-VL collapse into non-star geometries, revealing unstable procedural logic. GPT-4o and Claude produce coherent crochet-pattern-like text and correctly capture local yarn colors, yet fundamentally misinterpret the global motif: GPT-4o reconstructs a four-point star and begins with the brown yarn instead of blue, while Claude generates an eight-point motif rather than the intended seven. Gemini most accurately identifies the seven-point structure and selects plausible constructs such as bobbles for the star tips, but structural inconsistencies remain and yield visibly distorted shapes. These examples demonstrate that models can generate fluent, crochet-like descriptions while failing to preserve the algorithmic structure required for faithful pattern synthesis.

**Early-step instability reveals limits of procedural reasoning.** Figure 5 shows step-level results on Task D. Valid Pattern Rate increases with pattern depth but remains low overall: most models achieve under 15% validity in the first two steps, improve modestly in steps 3–4, and reach only 55–65% in later steps. This pattern reflects the difficulty of the initial steps, which must correctly initialize the program state such as defining stitch variables and maintaining balanced grouping. Errors made early propagate irreversibly, and later correctness often occurs only when the initial state is accidentally valid, indicating reliance on continuation heuristics rather than genuine procedural understanding. Larger models do not consistently perform better:

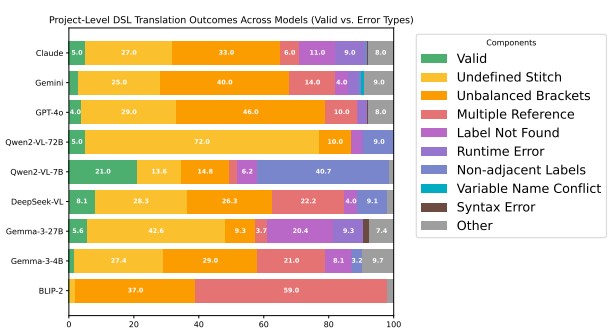

Figure 6: **Distribution of project-level DSL translation outcomes for each model, broken down into valid outputs and error categories.** Across all models, invalid programs dominate, with most failures arising from undefined stitches, unbalanced brackets, and multiple-reference errors. The wide spread of error types further illustrates the difficulty of maintaining global consistency and symbolic correctness when generating full crochet programs.

Qwen2-VL-72B underperforms Qwen2-VL-7B, and Gemma-27B underperforms Gemma-4B, suggesting that increased capacity improves descriptive fluency more readily than symbolic stability, and that scaling alone is insufficient for grammar-sensitive procedural tasks.

**Project-level synthesis exposes severe structural weaknesses.** Figure 6 further demonstrates the fragility of model performance when generating full CrochetPARADE programs. Valid outputs are exceedingly rare: even the strongest systems (Claude, Gemini, Qwen2-VL-7B) produce only 5–8% executable programs, while most others fall below 3%. The dominant failure modes (undefined stitches and unbalanced brackets) reflect unstable control over the DSL's vocabulary and grouping structure, and many models also exhibit multiple references, non-adjacent labels, and runtime errors. These error profiles indicate that models struggle to maintain consistent state and long-range structural dependencies across an entire pattern.

**Image-based similarity confirms lack of global structural fidelity.** Compilation verifies syntactic and structural correctness but cannot determine whether two DSL programs are semantically equivalent. To address this gap, we compute DINO similarity between the target crochet product image and the rendering produced from each model's executable program (valid outputs only). Figure 7 shows that similarity scores remain uniformly low across all models (0.10–0.17), far below the typical threshold for visually matched crochet images. Even when a model produces a compil-

able DSL program, the resulting rendering generally bears little resemblance to the intended pattern, indicating that syntactic validity does not imply correct procedural structure. The consistently low similarities reinforce that current multimodal LLMs fail to capture the global geometry and layout required for visually faithful crochet synthesis.

## 6 LIMITATIONS AND FUTURE WORK

Future improvements to Crochet-Bench span both dataset construction and modeling methodology. On the dataset side, CrochetBench currently relies on single product images and written instructions; extending the benchmark to multi-view and video settings would better capture aspects of crocheting that depend on motion, perspective, and temporal sequencing. The Crochet-PARADE DSL models the core of common crochet operations, but expanding it to cover additional construction techniques, advanced stitch types, and designer-specific conventions would broaden the range of patterns the benchmark can support. Incorporating richer supervision, such as expert ratings, correction traces, or human-verified program variants, would further strengthen evaluation in cases where multiple procedurally valid solutions exist.

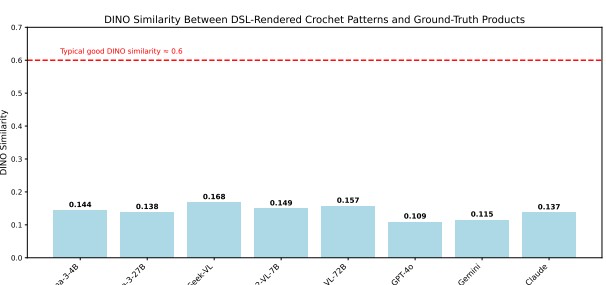

Figure 7: **Task D Project-level translation** results evaluating with **DINO similarity** between ground-truth images and DSL-rendered outputs generated from each model's DSL program (valid executable portion only). The red line marks an approximate "good" similarity threshold. All models fall well below this level, indicating that even executable DSL programs rarely reproduce the correct visual structure of the intended crochet design.

On the modeling side, our results highlight the need for architectures that go beyond visual recognition and text generation to support explicit state tracking, consistent counting logic, and long-range structural planning. Approaches that combine neural perception with symbolic scaffolds or memory-augmented components may help mitigate the drift and instability observed in DSL translation. Multimodal pretraining that includes procedural and topological data—such as assembly instructions, instructional videos, or structured manipulation tasks—may also narrow the gap between natural-language descriptions and executable program synthesis. Evaluation can likewise be expanded through hybrid pipelines that pair compilation checks with image-based comparisons of rendered outputs, providing complementary views of structural and perceptual fidelity.

More broadly, casting crochet as a program-synthesis task opens connections to established work in domain-specific languages for knitting, graphics, and robotics. This perspective naturally aligns with **CAD/CAM** workflows used in industrial crochet and warp-knitting machines, where pattern designs are compiled into machine-executable instructions. CrochetPARADE could serve as a standardized intermediate representation for such pipelines, bridging human-authored patterns with automated manufacturing systems. Finally, CrochetBench offers a platform for exploring neuro-symbolic approaches that integrate visual grounding with symbolic reasoning, aiming toward models that can generate procedures that are not only fluent, but also structurally correct and reliably executable.

## 7 CONCLUSION

CrochetBench provides a structured benchmark for assessing whether multimodal LLMs can move from recognizing visual content to executing the step-by-step procedures required to produce a crochet pattern. Across all four tasks, models demonstrate a consistent gap: they can identify stitches and retrieve plausible instructions, but they fail to generate structurally valid procedures or produce executable programs that match the intended design. Even when compilation succeeds, rendered outputs rarely capture the correct global geometry, revealing weaknesses in state tracking, counting logic, and long-horizon structural planning.

ETHICS STATEMENT

We acknowledge that the original crochet pattern PDFs are protected under copyright and therefore do not distribute raw files or full texts. Instead, we release only structured JSON annotations generated with GPT, reference URLs to the original sources, and our parsing and annotation scripts. The benchmark is provided strictly for non-commercial academic use. This approach enables reproducible research while respecting intellectual property and ensuring that our dataset serves as a tool for studying structured generation rather than redistributing creative works.

REPRODUCIBILITY STATEMENT

We have taken several steps to ensure the reproducibility of our results. All datasets, task templates, and evaluation procedures are documented in the main text and appendix. An anonymous repository containing the full source code, experiment scripts, and detailed reproduction instructions has been made publicly available at: `https://anonymous.4open.science/r/crochet-82E6/README.md`. This ensures that all reported results can be independently verified and extended by the research community.

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

# A  CROCHET PRIMER

Crochet patterns describe how to construct a textile artifact through a sequence of symbolic stitch instructions. Each instruction specifies an operation performed with a hook and yarn, and the resulting pattern is defined by the order, repetition, and spatial arrangement of these stitches. This appendix summarizes only the conventions needed to interpret the examples in our benchmark.

**Basic stitch types.**  Crochet relies on a small vocabulary of atomic stitches, each producing a loop with a characteristic height and structure. The most common stitches in U.S. notation are:

- **ch** (chain): foundational stitch used to begin rows or rounds.
- **sc** (single crochet): a short, dense stitch.
- **hdc** (half double crochet) and **dc** (double crochet): taller stitches that build height more quickly.
- **sl st** (slip stitch): a joining stitch used for connecting motifs or closing rounds.
- **bobble** (bobble stitch): a cluster of 5 partially completed double crochet stitches closed together into a single stitch.

These stitches can be combined in rows (worked back and forth) or rounds (worked in a circle).

**Pattern syntax and structure.**  Crochet instructions follow a compact symbolic notation. A pattern is organized into *rows* or *rounds*, each specifying a sequence of stitches. For example:

```
Row 3:  Ch 1, sc in each st across, turn.
```

Instructions may include:

- **Repetition**: indicated by parentheses and a multiplier, e.g., `(sc, ch 1) 3 times`.
- **Increases/decreases**: e.g., `2 sc in next st` (increase) or `sc2tog` (single-crochet two stitches together; decrease).
- **Stitch counts**: patterns often end rows or rounds with "—*N* sc," indicating the number of stitches that should remain.

**Relationship to symbolic representations.**  Each crochet instruction corresponds to a local modification of the fabric's topology. This makes crochet patterns naturally suited to symbolic or program-like representations such as CrochetPARADE, which encode stitches as structured primitives with explicit control flow (loops, groups, labels). Because stitch sequences fully determine the geometry of the final artifact, correctness can be assessed by verifying the structure of the generated program or by rendering the corresponding stitch graph.

This primer covers the minimal terminology required to interpret our dataset and evaluation tasks. For readers interested in additional background, standard crochet references provide extended stitch catalogs and diagram conventions.

# B  ADDITIONAL DATASET STATISTICS

## B.1  INSTRUCTION COMPLEXITY BY SKILL LEVEL

| Skill Level | Avg. Length | Median Length | Avg. Abbr. | Count |
|---|---|---|---|---|
| Beginner | 1,674 | 1,365 | 9.2 | 465 |
| Easy | 2,761 | 2,182 | 10.8 | 3,569 |
| Intermediate | 4,221 | 3,387 | 10.7 | 1,967 |
| Experienced | 7,689 | 6,729 | 9.8 | 80 |

Table 4: Instruction complexity by skill level. Length is measured in characters.

### B.2 Example Dataset Entry

### B.3 Skill Level Distribution

#### B.3.1 Overall Distribution

Table 6 summarizes the overall distribution of skill levels across the CrochetBench dataset. The majority of patterns are labeled as *easy* (58.7%), followed by *intermediate* (32.3%). Only a small fraction are classified as *beginner* (7.6%) or *experienced* (1.3%).[2]

One pattern (0.02%) is missing an annotated skill level.

#### B.3.2 Distribution by Project Type

We further break down skill levels by the top 10 most common project types. Results are shown in Table 7. In most categories, *easy* patterns dominate, typically ranging between 53–70%. *Intermediate* is the second most common, while *beginner* and *experienced* remain consistently low across categories.

Overall, the predominance of *easy* patterns reflects the accessibility of crochet as a craft and aligns with the goal of many project types to cater to a wide audience. The relative scarcity of *experienced*-level patterns suggests that most published resources emphasize broad usability rather than advanced expertise.

### B.4 Pattern Complexity Analysis

#### B.4.1 Instruction Length Statistics

We first analyze the distribution of instruction lengths, measured in raw character counts. As shown in Table 8, the average instruction length is over 3,200 characters, while the median is substantially lower at 2,453 characters, reflecting a long-tailed distribution. The most complex patterns extend beyond 30,000 characters, while some very short patterns are as small as 20 characters.

Out of 6,085 total patterns, 6,084 (99.98%) contain full instructions.

#### B.4.2 Abbreviation Statistics

Abbreviations, such as `sc`, `dc`, and `hdc`, are a distinctive element of crochet instructions. Table 9 reports abbreviation counts across all patterns. Most patterns contain about 10 abbreviations, with values ranging from 1 to 31.

#### B.4.3 Complexity by Skill Level

Instruction length correlates with the designated skill level. As shown in Table 10, beginner-level patterns average under 2,000 characters, while intermediate patterns extend to over 4,200. Experienced patterns are the longest, averaging 7,689 characters. Rare categories such as `easy to intermediate` skew extremely long due to outliers.

#### B.4.4 Most and Least Complex Project Types

Finally, we identify the most complex and simplest project types by average instruction length. Tables 11 and 12 list the top 10 categories. Garments such as dresses, vests, pants, and tunics are the most demanding, with average instructions exceeding 5,800 characters. By contrast, smaller accessories such as cowls, washcloths, scarves, and headbands are substantially shorter, typically under 2,000 characters.

Taken together, these results highlight strong alignment between project type, designated skill level, and instruction length. Garment-oriented projects require substantially longer and more complex instructions, while accessories and small decorative items remain simple and concise.

---

[2]Three additional rare labels were observed: `easy to intermediate` (1 pattern), `beginners` (1 pattern), and `beginner/easy` (1 pattern). Together they account for $< 0.1\%$ of the dataset.

## C    PROMPTS

### C.1    TASK A: STITCH RECOGNITION PROMPT

This task evaluates a model's ability to identify stitches present in a crochet product image.

---
**Stitch Recognition Prompt (Rendered Example)**

**SYSTEM PROMPT** You are a crochet stitch expert.
Given an image of a crochet product, identify all stitches that appear.
Requirements:
- Use only standard U.S. crochet abbreviations
(e.g., sc, hdc, dc, tr, ch, sl st, pop, etc.).
- Output must be a comma-separated list of abbreviations.
- Do not include explanations, extra text, or formatting beyond the list.

**USER PROMPT** Look at this crochet product image and list the stitches used.
[Image]

---

### C.2    TASK B: INSTRUCTION SELECTION PROMPT

This task evaluates a model's ability to choose the correct instructions from multiple-choice options.

---
**Instruction Selection Prompt (Rendered Example)**

**SYSTEM PROMPT**
You are a crochet expert. Your task is to determine which of the given options (A, B, C, or D) contains the correct crochet instructions for the image shown.

**USER PROMPT**
Look at this crochet image and choose which option best matches the instructions for making it.
[Image]
Options: {options text}
Choose exactly ONE option. Your answer should be only one letter: A, B, C, or D.

---

### C.3    TASK C: INSTRUCTION GENERATION PROMPT

This task evaluates a model's ability to generate complete crochet instructions from an image.

---
**Instruction Generation Prompt (Rendered Example)**

**SYSTEM PROMPT**
You are a professional crochet pattern writer. Examine the image of the finished crochet product carefully. Write a complete set of crochet instructions in the standard style used in published patterns.
Requirements:
- Use standard abbreviations: sc (single crochet), hdc (half double crochet),
dc (double crochet), tr (treble), ch (chain), sl st (slip stitch), rep (repeat).
- Organize the instructions row by row or round by round (e.g., "Rnd 1: ...", "Row 2: ...").
- If color changes are visible in the image, include them in the pattern.
- Keep the instructions concise and precise, as if for experienced crocheters.
- Output only the crochet pattern. Do not add any explanations, commentary, or extra text.

**USER PROMPT**
Generate step-by-step crochet instructions for this image.
[Image]

---

## C.4  TASK D (STEP-LEVEL): NL → DSL TRANSLATION PROMPT

This task evaluates whether a model can translate a single natural language instruction into exactly one line of compilable **CrochetPARADE** DSL code.

---

**Step-level NL → DSL Translation Prompt (Rendered Example)**

**SYSTEM PROMPT**
You are a crochet compiler. Translate the next instruction NL into one line of CrochetPA-RADE DSL.
Use consistent naming and syntax.
Important rules for translations:
1. Make sure your output ONLY contains the DSL code, nothing else.
2. Use the previous examples to understand the pattern of translation.
3. Be consistent in naming conventions with the examples.
4. Your output should be exactly one line of DSL code.

**USER PROMPT**
Now translate the NL into DSL:
NL:
DSL:

---

## C.5  TASK D (PROJECT-LEVEL): NL → DSL TRANSLATION PROMPT

This task evaluates whether a model can convert natural language crochet instructions (with optional images) into compilable CrochetPARADE DSL code.

---

**NL → DSL Translation Prompt (Rendered Example)**

**SYSTEM PROMPT**
You are a professional crochet pattern writer. Convert instructions + images into compilable CrochetPARADE DSL code. Output only the DSL code. No explanations, commentary, or extra text.
Example 1:
"image path":  `https://www.yarnspirations.com/cdn/shop/files/` `BRC0116-035467M.jpg`,

INSTRUCTIONS
  Note: Join with sl st to first sc at end of each rnd.
  Ch 2.
  **Rnd 1:** 6 sc in 2nd ch from hook. Join. (6 sc)
  **Rnd 2:** Ch 1. 2 sc in each sc around. Join. (12 sc)
  **Rnd 3:** Ch 1. (2 sc in next sc, 1 sc in next sc) repeat around. End with 1 sc. Join. (18 sc)
  **Rnd 4:** Ch 1. (2 sc in next sc, 1 sc in each of next 2 sc) repeat. End with 1 sc in last 2 sc. Join. (24 sc)
  **Rnd 5:** Ch 1. Sc in each sc around. Join. (24 sc)
  **Rnd 6:** Ch 1. (2 sc in next sc, 1 sc in each of next 3 sc) repeat. End with 1 sc in last 3 sc. Join. (30 sc)
  **Rnds 7–8:** Repeat Rnd 5 (sc in each sc). Join. (30 sc each round)
  **Rnd 9:** Ch 1. **Working in back loops only**: (2 sc in next sc, 1 sc in each of next 2 sc) repeat. End with 1 sc in last 2 sc. Join. (40 sc)
  **Rnd 10:** Ch 1. Sc in each sc around (both loops). Join. (40 sc)
  **Rnd 11:** Ch 1. (2 sc in next sc, 1 sc in each of next 3 sc) repeat. End with 1 sc in last 3 sc. Join. (50 sc)
  **Finish:** Fasten off.

DSL

---

```
¶ch.B
¶sc@B.A,5sc@B,ss@A
¶ch.A,sk,6sc2inc,ss@A
¶ch.A,sk,[sc2inc,sc]*6,ss@A
¶ch.A,sk,[sc2inc,2sc]*6,ss@A
¶ch.A,sk,24sc,ss@A
¶ch.A,sk,[sc2inc,3sc]*6,ss@A
¶[[ch.A,sk,30sc,ss@A
¶]]*2
¶ch.A,sk,[scbl,scbl@[@],2scbl]*10,ss@A
¶ch.A,sk,40sc,ss@A
¶ch.A,sk,[sc2inc,3sc]*10,ss@A
```

**USER PROMPT**
Now generate DSL code for the following:

[Image]

[Instructions]
Rnd 1: Ch 2, 6 sc in ring
Rnd 2: 2 sc in each (12)
Rnd 3: [Sc, sc, inc] around (16)
Rnd 4: [Tr, sc] repeat around

[DSL]

## D    CROCHETPARADE: PATTERN RENDERER, ANALYZER, AND DEBUGGER

**CrochetPARADE** (short for *Crochet Pattern Renderer, Analyzer, and Debugger*) is an interactive platform that enables users to author, visualize, test, and export crochet patterns in both 2D and 3D (Tassev, 2025). By combining a custom pattern grammar with simulation and rendering tools, CrochetPARADE addresses common issues of ambiguity, correctness, and interpretability in textual crochet instructions.[3]

**Core Capabilities.**

- **Interactive authoring and rendering.** Users write pattern instructions in the CrochetPA-RADE grammar and then invoke a "calculate" operation to convert those instructions into a virtual model. The system supports both 2D and 3D views, along with interactive controls such as zoom, rotation, and stitch highlighting.

- **Validation and debugging.** CrochetPARADE parses the input, checks for syntactic and consistency errors (e.g., mismatched stitch counts, impossible attachments), and flags over- or under-stretched stitches.

- **Export and interoperability.** From a rendered pattern, users can export:
  - A standard crochet chart (SVG) with conventional stitch symbols and labeled stitch connections.
  - A 3D model (GLTF format) for integration into external tools such as Blender.
  - The underlying pattern instructions text (in the CrochetPARADE grammar), ensuring reproducibility and sharing.

**Design Ideals and Rationale.**    CrochetPARADE is built to meet several design goals: (i) *unambiguous precision*, where the grammar is far more strict than free-form natural language, reducing

---

[3]https://www.crochetparade.org/

interpretive errors; (ii) *local computation*, since all parsing, simulation, and rendering occur client-side in the browser with no user instructions sent to a central server; and (iii) *open source extensibility*, as the platform is released under GPLv3, with the grammar manual provided under a Creative Commons BY-NC-SA license.

**Role in Our Work.** Within the context of CrochetBench, CrochetPARADE provides a rigorous target representation: model predictions can be compiled into CrochetPARADE instructions, validated for syntactic and structural correctness, and then visualized or executed. This enables evaluation beyond surface-level metrics (e.g., BLEU, ROUGE) toward *executor correctness*—whether a generated pattern is valid, renderable, and stitch-balanced.

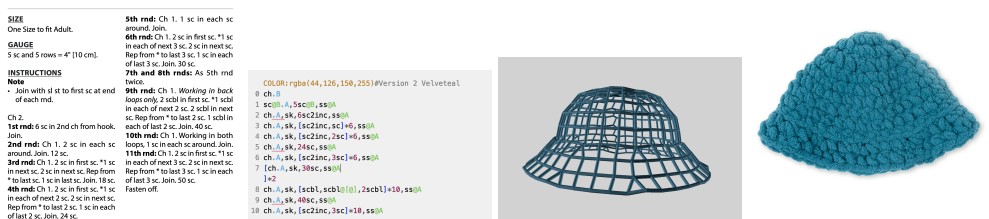

Figure 8: Example of the CrochetBench translation pipeline. (Left) Natural language crochet instructions from the dataset. (Second) Automatically translated into CrochetPARADE DSL, a formal stitch grammar. (Third) Mesh rendering generated from the DSL. (Right) Target crocheted item image provided in the dataset. This pipeline enables direct text-to-image consistency checks, automated validation, and future training of NL → DSL models, analogous to text-to-code generation.

# E    DSL ERROR TAXONOMY

To better understand failure cases in Task D, we extend the validator's error analysis with detailed subcategories and examples.

**Unbalanced Brackets.** Missing opening/closing parentheses or brackets.

> **Examples**
>
> ```
> Unbalanced brackets:   (sc,hc5,sltr)infl)
> ```

**Multiple References.** Improper formatting of references.

> **Example**
>
> ```
> Multiple references defined without parenthesis:
> (21ch),turn
> sk,(20sc)
> (2ndrow):Ch1.(1scbl)ineachchtoendofrow.Turn
> ```

**Undefined Stitch.** Undefined stitch types not in the dictionary.

> **Examples**
>
> ```
> ch1, ch3, scfp, hdc_bar
> ```

**Variable Naming Conflict.** Conflicts between variable names and stitch names.

> **Example**
>
> Error: variable name matches stitch name. For
> example, $ch=0$ cannot be used since 'ch' is a stitch
> name.

**Label Not Found.** Reference to a non-existent label.

> **Example**
>
> Label not found: C

**Non-Adjacent Labels.** Same label used for non-adjacent stitches.

> **Example**
>
> Cannot use same label over non-adjacent stitches.
> Consider using different labels.

**Turning Issue.** Misplaced turning commands.

> **Example**
>
> Turning can happen only at the end of a row.

**Runtime Errors.** Low-level parsing failures from the JavaScript compiler.

> **Examples**
>
> Cannot read properties of null (reading '0')
> Cannot use 'in' operator to search for 'attach_id' in
> NaN

**Multiplier Issue.** Improper formatting in multiplier.

> **Examples**
>
> Error: Exception during pattern parsing: Multiplier
> set, but no stitch found: ch.B

Table 5: Representative pattern entry from **CrochetBench**.

| Field | Value |
| --- | --- |
| Pattern Name | SKULL TRICK OR TREAT BAG (TO CROCHET) |
| Skill Level | Intermediate |
| Project Type | Bags or Purses |
| Measurements | 15 cm diameter × 15 cm high (excluding handle) |
| Gauge | 13 sc and 14 rows = 10 cm |
| Materials | Lily® Sugar'n Cream (White, Black), 5 mm hook, cardboard |
| Image | `https://www.yarnspirations.com/cdn/shop/products/SCC0303-005314M.jpg` |
| Source | `input_file/Bags+Purses/SCC0303-005314M.pdf` |
| Instructions | **Instructions:** |
| | Note: Ch 2 at beg of each rnd counts as hdc. |
| | **BAG** |
| | With MC, ch 4. Join with sl st to form ring. |
| | 1st rnd: Ch 2. 11 hdc in ring. Join with sl st to top of ch 2. 12 hdc. |
| | 2nd rnd: Ch 2. 1 hdc in same sp as sl st. 2 hdc in each hdc around. Join. 24 hdc. |
| | 3rd rnd: Ch 2. 1 hdc in same sp. 1 hdc in next hdc. *2 hdc in next hdc, 1 hdc in next.* Rep around. Join. 36 hdc. |
| | 4th rnd: Ch 2. 1 hdc in each hdc around. Join. |
| | 5th rnd: Ch 2. 1 hdc in same sp. 1 hdc in next 2 hdc. *2 hdc, 1 hdc in next 2.* Join. 48 hdc. |
| | 6th rnd: As 4th rnd. |
| | 7th rnd: Ch 2. 1 hdc in next 2 hdc. *2 hdc, 1 hdc in next 3.* Rep. 60 hdc. |
| | 8th rnd: Ch 2. Back loops only, 1 hdc around. Join. |
| | 9th–13th rnds: Ch 2. 1 hdc in each hdc around. Join. |
| | 14th rnd: Ch 2. 1 hdc in same sp. 1 hdc in next 4 hdc. *2 hdc, 1 hdc in next 4.* Join. 72 hdc. |
| | 15th rnd: Ch 2. 1 hdc in same sp. 1 hdc in next 5 hdc. *2 hdc, 1 hdc in next 5.* Join. 84 hdc. |
| | 16th–22nd rnds: Ch 2. 1 hdc in each hdc around. Join. |
| | 23rd rnd: Ch 2. 1 hdc in next 4 hdc. *Hdc2tog, 1 hdc in next 5.* Rep. Hdc2tog. Join. 72 sts. |
| | 24th rnd: Ch 2. 1 hdc in next 3 hdc. *Hdc2tog, 1 hdc in next 4.* Rep. Join. 60 sts. |
| | 25th rnd: Ch 2. 1 hdc in next 2 hdc. *Hdc2tog, 1 hdc in next 3.* Rep. Join. 48 sts. Fasten off. |
| | **Eyes (Make 2)** |
| | With A, ch 8. |
| | 1st rnd: 2 sc in 2nd ch from hook. 1 sc in next 5 ch. 3 sc in last ch. Continue on rem loops, 1 sc in each ch. Join. 17 sc. |
| | 2nd rnd: Ch 1. 3 sc in first sc. 1 sc in next 7 sc. 3 sc in next sc. 1 sc in next 8 sc. Join. Fasten off. |
| | **Handle** |
| | With MC, ch 45. |
| | 1st row: 1 sc in 2nd ch from hook. 1 sc across. 44 sc. Turn. |
| | 2nd row: Ch 1. 1 sc across. Turn. |
| | Rep last row 4 more times. Fasten off. |
| | **Finishing** |
| | Sew Eyes to Bag. Embroider mouth and teeth with A. Attach Handle. Cut cardboard circle to fit bottom. |

Table 6: Overall skill level distribution. Percentages are relative to all patterns with annotated skill levels.

| Skill Level | Count | Percentage |
|---|---|---|
| Easy | 3569 | 58.66% |
| Intermediate | 1967 | 32.33% |
| Beginner | 465 | 7.64% |
| Experienced | 80 | 1.31% |
| **Total** | **6084** | **100%** |

Table 7: Skill level distribution by top 10 project types. Percentages are within each project category.

| Project Type | Easy | Intermediate | Beginner | Experienced |
|---|---|---|---|---|
| Afghans & Blankets | 56.1% | 35.3% | 7.0% | 1.5% |
| Hats | 61.3% | 27.8% | 10.1% | 0.7% |
| Sweaters & Cardigans | 56.6% | 35.9% | 5.0% | 2.5% |
| Shawls & Wraps | 52.7% | 41.8% | 4.2% | 1.2% |
| Scarves | 63.2% | 20.7% | 16.1% | – |
| Pillows & Poufs | 70.0% | 22.9% | 6.5% | 0.7% |
| Amigurumi & Toys | 64.0% | 33.2% | 2.1% | 0.7% |
| Bags & Purses | 53.8% | 39.0% | 6.8% | 0.4% |
| Décor | 58.4% | 33.3% | 6.5% | 1.7% |
| Dishcloths | 62.6% | 27.5% | 9.9% | – |

Table 8: Instruction length statistics (in characters).

| Statistic | Value |
|---|---|
| Average | 3216.0 |
| Median | 2453.0 |
| Min | 20 |
| Max | 30634 |
| 25th percentile | 1511.8 |
| 75th percentile | 4136.2 |
| 90th percentile | 6403.9 |

Table 9: Abbreviation count statistics.

| Statistic | Value |
|---|---|
| Average | 10.6 |
| Median | 10.0 |
| Min | 1 |
| Max | 31 |

Table 10: Instruction length and abbreviation counts by skill level.

| Skill Level | Avg. Length | Median Length | Avg. Abbr. | Count |
|---|---|---|---|---|
| Easy to intermediate | 13812.0 | 13812.0 | 21.0 | 1 |
| Experienced | 7689.4 | 6729.0 | 9.8 | 80 |
| Intermediate | 4221.3 | 3387.0 | 10.7 | 1967 |
| Easy | 2760.7 | 2182.0 | 10.8 | 3569 |
| Beginner | 1673.9 | 1365.0 | 9.2 | 465 |
| Beginners | 1633.0 | 1633.0 | 11.0 | 1 |
| Beginner/Easy | 1063.0 | 1063.0 | – | 1 |

Table 11: Top 10 most complex project types (by average instruction length).

| Project Type | Avg. Length | Median | Count |
|---|---|---|---|
| Dresses | 6484.9 | 5799.0 | 34 |
| Vests | 6032.0 | 5193.5 | 64 |
| Pants | 5866.7 | 5409.0 | 11 |
| Tunics | 5850.4 | 5832.0 | 29 |
| Sets | 5625.5 | 4847.0 | 111 |
| Sweaters & Cardigans | 5429.2 | 5113.0 | 357 |
| Amigurumi & Toys | 5322.4 | 4505.0 | 286 |
| Jackets | 5311.9 | 4831.0 | 31 |
| Onesies & Rompers | 5263.4 | 5181.0 | 5 |
| Aprons | 4467.8 | 4494.0 | 11 |

Table 12: Top 10 simplest project types (by average instruction length).

| Project Type | Avg. Length | Median | Count |
|---|---|---|---|
| Cowls | 1288.3 | 956.5 | 154 |
| Washcloths & Mitts | 1502.5 | 1420.0 | 28 |
| Scarves | 1567.3 | 1221.0 | 304 |
| Headbands | 1617.5 | 1475.5 | 38 |
| Dishcloths | 1688.4 | 1571.0 | 222 |
| Coasters | 1750.3 | 1625.0 | 26 |
| Booties | 1921.9 | 1938.5 | 24 |
| Jewelry | 1960.3 | 1549.0 | 55 |
| Super Scarves | 2007.6 | 1213.0 | 13 |
| Tech Accessories | 2011.1 | 2099.0 | 13 |

